# Recent Developments in the Photocatalytic Treatment of Cyanide Wastewater: An Approach to Remediation and Recovery of Metals

**Luis Andrés Betancourt-Buitrago** [1], **Aracely Hernandez-Ramirez** [2],
**Jose Angel Colina-Marquez** [3,*], **Ciro Fernando Bustillo-Lecompte** [4], **Lars Rehmann** [5] and
**Fiderman Machuca-Martinez** [1]

1   Escuela de Ingeniería Química, Universidad del Valle, Calle 13 #100-00. Cali A.A. 25360, Colombia;
    betancourt.luis@correounivalle.edu.co (L.A.B.-B.); fiderman.machuca@correounivalle.edu.co (F.M.-M.)
2   Facultad de Ciencias Químicas, Universidad Autónoma de Nuevo León, San Nicolás de los Garza 64570,
    Mexico; aracely.hernandezrm@uanl.edu.mx
3   Department of Chemical Engineering, Universidad de Cartagena, Sede Piedra de Bolívar, Avenida del
    Consulado 48-152, Cartagena A.A. 130001, Colombia
4   Graduate Programs in Environmental Applied Science and Management, and School of Occupational and
    Public Health, Ryerson University, 350 Victoria Street, Toronto, ON M5B 2K3, Canada;
    ciro.lecompte@ryerson.ca
5   Department of Chemical and Biochemical Engineering, Thompson Engineering Building, Western
    University, London, ON N6A 5B9, Canada; lrehmann@uwo.ca
*   Correspondence: jcolinam@unicartagena.edu.co

**Abstract:** For gold extraction, the most used extraction technique is the Merrill-Crow process, which uses lixiviants as sodium or potassium cyanide for gold leaching at alkaline conditions. The cyanide ion has an affinity not only for gold and silver, but for other metals in the ores, such as Al, Fe, Cu, Ni, Zn, and other toxic metals like Hg, As, Cr, Co, Pb, Sn, and Mn. After the extraction stage, the resulting wastewater is concentrated at alkaline conditions with concentrations up to 1000 ppm of metals. Photocatalysis is an advanced oxidation process (AOP) able to generate a photoreaction in the solid surface of a semiconductor activated by light. Although it is well known that photocatalytic processes can remove metals in solution, there are no compilations about the researches on photocatalytic removal of metals in wastewater with cyanide. Hence, this review comprises the existing applications of photocatalytic processes to remove metal and in some cases recover cyanide from recalcitrant wastewater from gold extraction. The use of this process, in general, requires the addition of several scavengers in order to force the mechanism to a pathway where the electrons can be transferred to the metal-cyanide matrices, or elsewhere the entire metallic cyanocomplex can be degraded by an oxidative pathway.

**Keywords:** UV-LED; photoreactors; mining wastewater; cyanide; metal removal

## 1. Introduction

Gold has always had a high value since prehistoric times as ornaments in rituals, and it occupies an essential role in the world economy. By mid-2017, the world gold reserves were around 33,450 metric tons, with a demand of 4337 tons in 2016, destined for jewelry (47%), technology (7%), investments (36%), and central banks (9%) [1].

The gold exploitation depends on the way it is present in minerals, and its extraction can be done in the acid phase (pH < 3) with thiourea, thiocyanate, chlorine, aqua regia, ferric chloride; in neutral

phase with thiosulfate, halogens, sulfuric acids, bacteria; and in alkaline phase (pH > 10) with cyanide, ammonium cyanide, ammonium, sulfur, and nitriles [2,3]. However, the practical application of these processes is limited to extraction in the alkaline phase using cyanide because of its high selectivity with respect to gold [4–6].

Latin America and the developing countries exhibit one of the primary gold and silver exploitation scenarios based on the leaching of ores with solvents, such as sodium cyanide (NaCN)—The Merrill-Crowe Process. In this extraction process, the gold-concentrated effluent is later taken to a precipitation stage with the use of zinc, called cementation [7]. The wastewater resulting from this process is rich in heavy and non-heavy metals, poor of gold, and it contains dissolved silver, which is very harmful to the environment [8]. The mining wastewater is well known to be the predominant cause of pollution problems in surface water bodies (lakes and rivers). The problems, such as death, due to poisoning, lead poisoning; cancer, due to chromium, blindness and congenital malformations, are attributed to the contamination of surface and underground water sources [9]. Furthermore, heavy metals in these waters could be bioaccumulated and present biomagnification causing serious health effects, due to their high levels of toxicity [10–12]. Besides, mining wastewaters can show problems of metal mobility and local cyanide release where they are stored; mining wastewaters are directly discharged to tailings ponds for periods of three to six months where degradation is expected by the sun (photolysis and evaporation) [13].

In large-scale operations, wastewater treatment is carried out with highly oxidizing processes, such as chlorination, sulfur dioxide, hypochlorite oxidation, electrolytic oxidation, ozonation, use of hydrogen peroxide, high thermal transformation, biological treatments, adsorption on activated carbon, among others, usually at high oxidation conditions and operation cost [14].On the other hand, advanced oxidation processes (AOPs) have the advantage of removing liquid and recalcitrant gaseous matrices by non-selective chemical species. Among these processes, heterogeneous photocatalysis (HPC) has received considerable attention as a promising technology, able to use renewable energy from the sun. It is conventionally defined as the acceleration of the rate of a chemical reaction, induced by the absorption of light by a catalyst or coexisting molecule. This definition of photocatalysis may be the most widely accepted as it encompasses all aspects of the field, including photosensitization [15].

HPC is one of the AOPs that allows the elimination of toxic compounds in a non-selective pathway, due to the generation of oxidizing species, such as the hydroxyl radical ($^{\bullet}OH$), perhydroxyl radical ($HOO^{\bullet}$), superoxide ($O_2^{\bullet-}$), and photogenerated holes (h+), transforming recalcitrant and toxic molecules into biodegradable or less harmful compounds [16]. Photocatalytic processes not only are applied to oxidize recalcitrant organic matter, but also to promote reduction reactions. Some examples are the photoreduction of benzaldehyde to benzyl alcohol, metallic ions, such as $Fe^{3+}$, $Cr^{6+}$, $Hg^{2+}$, $Cu^{2+}$, inorganic nitrogen and carbon dioxide to formic acid, simulating part of artificial photosynthesis.

The photocatalytic reduction represents an option when traditional oxidative pathways are not feasible and when the nature of semiconductor is able to transfer electrons at a high energy level on its conduction band. Although the current applications of large-scale photocatalytic processes are scarce, different assessments have been made for the contamination associated with gold mining wastewaters. In this review, different photocatalytic processes used for the elimination of synthetic and real cyanide matrices of gold extraction are explained and described.

## 2. Production and Characterization of Cyanide Wastewater

Cyanidation is used when gold is in the pyrite form and is not extractable by physical separation methods. This process is carried out through the use of sodium cyanide in the alkaline phase and with an excess of oxygen, as shown in Equation (1) [17]. Once Au is extracted from the ores, the gold is precipitated by adding Zn (cementation), replacing the gold of the aurocyanide ion with zinc cyanide and precipitating it in metallic form, as indicated in Equation (2). Although the efficiency of the process is in the order of 99%, the wastewater has metal cyano-complexes strong acid dissociable (SAD), such

as iron, copper, and cobalt, as well as weak acid dissociable (WAD), such as nickel, silver, zinc, and arsenic [14].

Leaching or cyanidation:

$$4Au + 8NaCN + O_2 + 2H_2O \rightarrow 4Na[Au(CN)_2] + 4NaOH \tag{1}$$

Cementation:

$$2NaAu(CN)_2 + 4NaCN + 2Zn + 2H_2O \rightarrow 2Na_2Zn(CN)_4 + 2Au\downarrow + \uparrow H_2 + 2NaOH \tag{2}$$

Table 1 shows the metallic and semi-metallic cyano-composites, sorted by the logarithms of their stability constants. Thus, the most unstable compound corresponds to the hydrogen cyanide in the gas phase; the easily dissociable WAD corresponds to complexes of Cd, Zn, Ag, Ni, Cu, Cr, and the most stable SAD correspond to complexes of Fe, Au, Co. The stability of strong complexes makes necessary the use of tailings ponds for removing them by solar-evaporation [18].

**Table 1.** Stability of cyano-metallic complexes [19].

| Group | Species | Toxicity [20] | Stability Constant (Log $K_n$) |
|---|---|---|---|
| Free cyanide | $CN^-$ | High | n.a. |
| | $HCN_{(g)}$ | | 9.2 |
| Simpler compounds: Easily soluble | NaCN, KCN, Ca(CN)$_2$, Hg(CN)$_2$, Zn(CN)$_2$, CuCN, Ni(CN)$_2$, AgCN | High | n.d. |
| Weak complex (*WAD—Weak Acid Dissociable*) | $Cd(CN)_4^{2-}$ | Intermediate | 17.9 |
| | $Cd(CN)_3^-$ | | n.d. |
| | $Zn(CN)_4^{2-}$ | | 19.6 |
| | $Ag(CN)_2^-$ | | 20.5 |
| | $Ni(CN)_4^{2-}$ | | 30.2 |
| | $Cu(CN)_3^{2-}$ | | 21.6 |
| | $Cr(CN)_6^{3-}$ | | n.a. |
| | $Cr(CN)_6^{3-}$ | | n.a. |
| Strong complexes (SAD—*Strong Acid Dissociable*) | $Fe(CN)_6^{4-}$ | | 35.4 |
| | $Fe(CN)_6^{3-}$ | Low | 43.6 |
| | $Au(CN)_2^-$ | | 38.3 |
| | $Co(CN)_6^{3-}$ | High | 64.0 |
| Unstable inorganic | $SCN^-$, $CNO^-$ | High | n.d. |
| Aliphatic organic | Acetonitrile, acrylonitrile, adiponitrile, propionitrile | Intermediate | n.d. |

On the other hand, weak complexes are easily hydrolyzable by changing the pH of the solution. In principle, weak complexes tend to be destroyed over three months with or without photolysis; however, strong complexes, such as $Fe(CN)_6^{3-}$, $Co(CN)_6^{3-}$ remain over time, turning these waters into recalcitrant. Additionally, degradation products, such as $NH_3/NH_4^+$, $NO_2^-$, $NO_3^-$, $CNO^-$, sulfates, and carbonates are formed by the slow rupture of cyano-metallic complexes. Thus, the resulting wastewater (concentrated by these complexes) is not suitable for being poured into surface bodies of water [13,21].

## 3. Existing Treatment Options

The existing treatment options of oxidative processes for the treatment of cyanide wastewater, such as natural attenuation [18,22], chemical oxidation [23], thermal treatments, precipitation, biologic oxidation [24–27], and ionic adsorption [28], are well documented [18]. Nonetheless, strictly

photocatalytic treatments are scattered, and there is no clarity of existing photocatalytic processes applications for degradation of cyanide complexes.

Figure 1 shows a relational diagram constructed with VOS Viewer® using technology watch tools [29] on the main topics related to cyanide. In this figure, it is observed that the relationship with the word "photocatalysis" is not very broad. However, it appears related to "activated carbon" and "titanium dioxide" (in yellow). Likewise, other technologies appear, such as: "Biodegradation", "ozone", "hydrogen peroxide", "electroplating", and "adsorption". For this review, some applications of photocatalytic processes used in the degradation of this type of cyano-metallic wastewater are depicted.

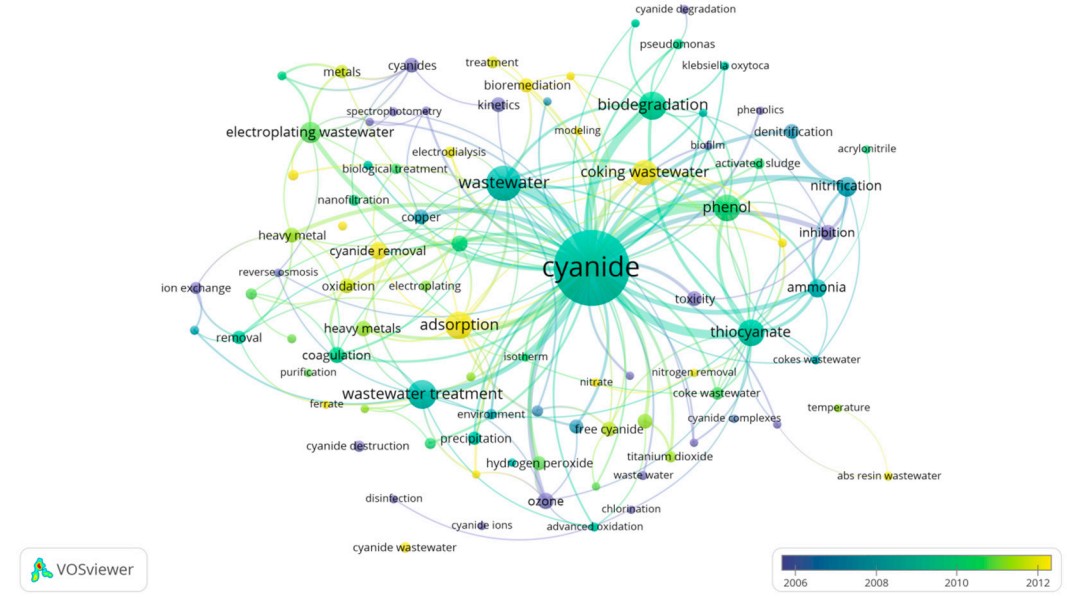

**Figure 1.** Keyword correlation map. Database: Web of Science. 464 documents. Date of consultation: 22 February 2018. Built with VOSViewer v1.6.7.

## 4. Photocatalytic Treatment Alternatives

### 4.1. Classic Oxidative Photocatalysis

AOPs are generally based on mechanisms capable of producing profound changes in the chemical structure of pollutants. Heterogeneous Photocatalysis (HPC) is a photochemical AOP where redox reactions are promoted by the interaction of a semiconductor catalyst and photons, generating active species able to degrade recalcitrant organic matter to allowable concentrations for final discharge [30].

Figure 2 shows the photocatalytic mechanism on a $TiO_2$ particle. The photocatalytic process initiates when a semiconductor is irradiated by photons (from solar or artificial light) whose energy is equal or greater than the band-gap energy (Eg) of the semiconductor, promoting the electrons from the balance band to the conduction band. Then, the electrons at the conduction band can react with the adsorbed oxygen on the semiconductor surface to form superoxide radicals ($O_2^{\bullet-}$), and other radicals with water, such as the perhydroxyl ($HOO^\bullet$) or generating reduction reactions by the direct or indirect transfer with adsorbed compounds. On the other hand, the holes in the balance band can react with the adsorbed water generating hydroxyl radicals ($^\bullet OH$), characteristics of the AOPs [31]. Conversely, it has also been evidenced that hydroxyl radicals are obtained by reduction of $H_2O_2$ and not by direct oxidation of water adsorbed on the catalyst surface [32]. Furthermore, the holes in the valence band can react directly with other adsorbed molecules as sacrificing agents oxidizing them to simpler substances [32]. This final step is the most argued mechanism in the photocatalytic treatment of recalcitrant molecules.

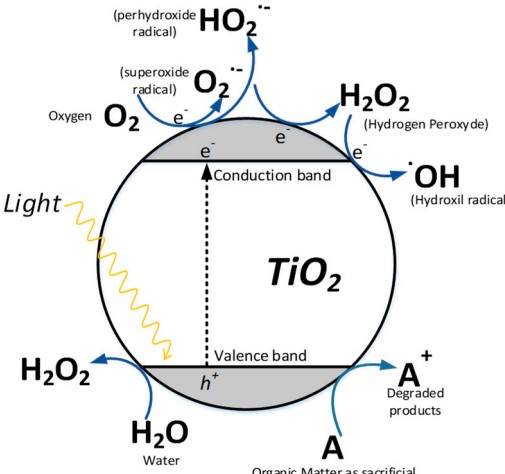

**Figure 2.** One electron reduction steps of oxygen to hydroxyl radical and two electron oxidation steps of water to $H_2O_2$. Adapted from Reference [32].

In the case of the cyanide, the use of different catalysts, such as titanium dioxide, nickel oxide, zinc oxide, platinum, zirconium, cadmium, and cobalt, have been studied for removing potassium cyanide and sodium cyanide [33–50].

In general, all the photocatalytic evaluations are carried out using the oxidative chemical pathway that works in the presence of oxygen. This process transforms the adsorbed substances in less toxic compounds (nitrates, carbonates, carbon dioxide, and nitrogen). Table 2 shows the most applied processes in matrices containing synthetic cyanide solutions of KCN and NaCN. It can be noticed that the initial concentrations oscillate around 100 ppm of $CN^-$ (3.2 mM $CN^-$). Moreover, photocatalytic degradation with doped $TiO_2$ and ZnO has been evaluated, with the main doping agents of Ce, Ag, Zn, Pt, and Co. These photocatalysts have achieved degradations of free cyanide (over 8 and 10%) in oxic conditions and with artificial radiation using UV-C (light emitting diodes) LED light. The photocatalytic evaluations in batch and continuous reactors for degradation of organic cyanide compounds show how they have degraded and mineralized acetonitrile ($CH_3CN$) in the gaseous and liquid phase, obtaining better results with the gas phase [47].

Other combinations for free cyanide involve combined treatments of oxidation processes, such as photocatalysis/ozone/electrolysis/electrocoagulation that achieved degradations greater than 90% [51–59]. Although in those photocatalytic evaluations, more than 90% of the substrate was removed. Most of them were carried out with synthetic solutions of pure free cyanide, which does not address the issue related to the metal-cyanide complex remediation at laboratory scale, as shown in Table 2.

The photocatalytic degradation of free cyanide is summarized in Figure 3. As mentioned before, the oxidation is carried out by different pathways: Oxidation by holes ($h^+$), oxidation by superoxide ($O_2^{\bullet-}$) and oxidation by hydroxyl radicals ($^{\bullet}OH$). The reaction of $CN^-$ with radicals transforms it into cyanate ($CNO^-$), ammonium ($NH_4^+$), nitrates ($NO_2^-$, $NO_3^-$) and carbonates ($HCO_3^-$, $CO_3^{2-}$). Although these photocatalytic treatments were used for the degradation of free cyanide, all applications were limited to synthetic solutions. Complexing agents like the metals in the ores represent the main problem of mining wastewater.

**Table 2.** Photocatalytic treatments applied to free cyanide matrices.

| Year | Substance [$C_0$] | Source of Light | Wavelength | Type of Reactor | Degradation/Reaction Time | Catalyst | Main Findings |
|---|---|---|---|---|---|---|---|
| 1992 | KCN [100 ppm] | 14 W UV Hg Low Pressure | 360 nm | Compact Square batch reactor | 100%/60 min | $TiO_2$ P25 | Achieve total degradation to nitrates and cyanates. They find the $CO_2$ in air bubbling as harmful for the photocatalytic mechanism [33] |
| 1999 | NaCN, NaCNO [3.85 mM] | Solar light | Solar spectrum | CPC pilot scale | 100%/4.1 Einstein accumulated | $TiO_2$ P25 | Total degradation with solar light, but kinetics is only related to accumulated energy [34] |
| 2001 | NaCN [666 ppm $CN^-$] | 450 W, 700 W Hg high-pressure lamp | UV-A | Laboratory Batch | 1.5 mmol/h $H_2$ produced at 70 °C and 700 W | $NiO/TiO_2$ | The process produced hydrogen and cyanate from cyanide as a photocatalytic strategy of remediation [50] |
| 2002 | Free Cyanide, phenol, atrazine, EPTC, dichloroacetic acid, and Cr(VI) among others. | Solar | Solar spectrum | Pilot-scale PSA–Solar platform of Almeria | 100%/N.D. | photo Fenton and photocatalysis applications. | Several experiments applied at a solar pilot plant in Almeria with successful results [60] |
| 2002 | KCN [100 ppm] | 150W Hg medium pressure lamp | >300 nm | Batch cylindrical | 47%/2 h $TiO_2$/SBA-15 | Supported $TiO_2$ on SBA-15 and MCM-41 | Achieved geometry optimization using the support SBA. However, degradation resulted low [43] |
| 2002 | NaCN [100 ppm $CN^-$] | 150W Hg Medium pressure | n.a. | Batch cylindrical | 50%/350 min | $TiO_2$ Sol-gel method on four different support | Achieved a low degradation of free cyanide exploring a novel geometry configuration on the $TiO_2$ distribution [48] |
| 2003 | NaCN [3.85 mM] | n.a. | n.a. | n.a. | 100%/420 min | $TiO_2$ P25 | Although total degradation was achieved, authors argue the photonic efficiency is very low and radical recombination occurred. They propose a very detail degradation kinetic mechanism [61] |
| 2004 | KCN [50 ppm] | 450 W High-pressure Hg lamp | >300 nm | binaural pyrex batch with intern lamp | n.a. | $TPA/TiO_2$, $Cs$-$TPA/TiO_2$ | They determine the interaction of $CN^-$ with holes and electrons photogenerated. The Cs resulted in photocatalytic inhibition [62]. |
| 2005 | $CH_3CN$ (gas and liquid) [24 mM] | 500 W Hg medium pressure lamp | 365 nm | Annular photoreactor steady state for liquid and gas phase | 21%/4 g gas phase 35%/5 g liquid phase | $TiO_2$ anatase for gas, and $TiO_2$ P25 for the liquid phase | Photocatalytic activity was low, and free cyanide ions remain in solution [47] |
| 2007 | $NH_3$, HCOOH, $CN^-$ from Electric Power Plant wastewater [10 ppm $CN^-$, 1700 HCOOH, 150 ppm $NH_3$] | 150 W Hg lamp | 190–280 nm | Batch cylindrical | 100% CN 90% $NH_3$ 100% HCOOH/10 min | $TiO_2$ P25 + $H_2O_2$ | Requires addition of $H_2O_2$ to enhance photocatalytic degradation [45] |

**Table 2.** *Cont.*

| Year | Substance [$C_0$] | Source of Light | Wavelength | Type of Reactor | Degradation/Reaction Time | Catalyst | Main Findings |
|---|---|---|---|---|---|---|---|
| 2007 | KCN [45 ppm $CN^-$] | 400 W Hg UV Lamp | >300 nm | Recirculating cylindrical photoreactor | 5%/100 min | Sol-gel $TiO_2/SiO_2$ | Apply an optimization methodology to optimize the photonic efficiency of the photoreactor. However, a very low photodegradation was evidenced [63] |
| 2007 | KCN [40 ppm $CN^-$] | 400 W Hg medium pressure lamp | >300 nm | Cylindrical with reflector | 95%/60 min | Three photocatalysts were evaluated: $TiO_2$ P25, DBH $TiO_2$, nanometric $TiO_2$. | Evaluated the photocatalytic degradation with three photocatalysts and with the addition of $O_3$. A good degradation was achieved but the addition of $O_3$ instead $O_2$ resulted in photocatalytic inhibition [59] |
| 2008 | KCN [3.85 nM] | 80 W and 36 W Low-pressure lamp | UV-A | Cylindrical photoreactor | n.a. | $TiO_2$ P25, $TiO_2/SiO_2$ | The authors proposed an intrinsic kinetic model of cyanide degradation with an accurate fitting of experimental data. The study was more kinetic than a photocatalytic evaluation [64] |
| 2008 | NaCN, gasification plant wastewater [10 ppm $CN^-$] | Solar light | 200 W/cm$^2$ of solar spectrum concentrated with a Fresnel Lens | Cylindrical photoreactor | 100%/90 min | $TiO_2$ P25. | The evaluated the effect of solar light using a Fresnel lens to concentrate energy. They required the addition of $H_2O_2$ to achieve total mineralization of free cyanide [46] |
| 2008 | KCNO, $Fe^{+4}$ [1 mM $CNO^-$] [1 mM $Fe^{+4}$] | n.a. | UV-A | Borosilicate glass cylindrical | 80% cyanate degradation/120 min | $TIO_2$ P25 | The process reduced ferrate(VI) and oxidated cyanate in a Fe(VI)-$TiO_2$-UV-$NCO^-$ system. However, the role of the $TiO_2$ in the degradation was not specified. The possible reduction-oxidation mechanism for $Fe^{+4}$ reduction was not clarified [65] |
| 2008 | KCN [100 ppm $CN^-$] | 15 W Hg low-pressure lamp | UV-A | Cylindrical batch | 100%/350 min | $TiO_2$ P25 | The degradation was done using 10.5 mM EDTA as a hole scavenger. Addition of EDTA evidenced an increase in the cyanide oxidation to $CNO^-$ [49] |
| 2009 | KCN [30 ppm $CN^-$] | 400 W and 36 W Blacklight lamp | 365 nm | Annular reactor | 100%/120 min | $TiO_2/SiO_2$ | They evaluate and compared the scaling-up process from laboratory to pilot plant, using supported $TiO_2$. Total elimination of cyanide was achieved in both systems. Propose a scaling up methodology for photoreactors [66] |

**Table 2.** *Cont.*

| Year | Substance [$C_0$] | Source of Light | Wavelength | Type of Reactor | Degradation/Reaction Time | Catalyst | Main Findings |
|------|-------------------|-----------------|------------|-----------------|---------------------------|----------|---------------|
| 2009 | NaCN [400 ppm] | 450 W Halide lamp | UV-A | Cylindrical glass batch | 90%/30 min | TiO$_2$ nanoparticles coupled with an electrocoagulation recovery | It is proposed a recovery technique using electrocoagulation after a typical photocatalytic cyanide degradation. A study of TiO$_2$ reuse was also performed [54] |
| 2010 | KCN [250 ppm] | 8 W Hg lamp | 365 nm | Batch cylindrical | 40%/100 min | Ce-ZnO sonochemical impregnation | Doping relations of 2% Ce-ZnO calcined at 500 °C. This photocatalyst works better in the visible region. There is an excess of light applied to the system, which could mix the photocatalysis with the photolysis effect on CN$^-$ degradation [35] |
| 2010 | KCN [11 mM CN$^-$] | 8 W Hg Lamp | 365 nm | Batch cylindrical Reactor | 86%/90 min | Ag-ZnO sonochemical impregnation synthesis | Ag-ZnO was found to be three times better than ZnO pure [36] |
| 2011 | KCN [10 mM] | 150 W halide and 8 W Hg UV lamp | 365 nm UV | Annular batch reactor | 16%/150 min | ZnO-TiO$_2$ | Photocatalytic activity was demonstrated but with important radiant field losses in the photoreactor [37] |
| 2013 | KCN [100 ppm] | 150 W fluorescent lamp | 450 nm | Batch cylindrical reactor | 98%/60 min | Pt-TiO$_2$- hydroxyapatite. Prepared by Sonic method. | Hydroxyapatite enhanced the photocatalytic behavior of bare suspended TiO$_2$ [38] |
| 2013 | KCN [100 ppm] | 150 W fluorescent lamp | 450 nm | Batch annular reactor | 100%/20 min | Pt/ZrO$_2$-SiO$_2$ prepared by a photo-assisted deposition method. | Evaluated the effect of catalyst load on the reactor [39] |
| 2014 | KCN [100 ppm] | 150 W fluorescent blue lamp | 450 nm | Batch cylindrical reactor | 100%/360 min 96%/240 min | Co-TiO$_2$-SiO$_2$ prepared by a photo-assisted method and impregnation. | Obtained the best catalyst load obtained at 0.08 g/L and a decreased in the TiO$_2$ band-gap with the total elimination of CN$^-$ [41] |
| 2015 | NaCN [30 ppm] | UV-LED | Not specified. UV-A UV-B UV-C | Submerged cylindrical LED photoreactor | 100%/>600 min | TiO$_2$ P25 | Demonstrated the possibility of using LED as a source of UV light in a photocatalytic treatment. The most efficient was UV-C, due to photolytic effect [21] |
| 2015 | KCN [100 ppm] | 500 W Xe bulb lamp | >420nm | Pyrex reaction cell | 100%/60 min | MWCNT/Au-TiO$_2$ | They found carbon nanotubes beneficial for photocatalytic degradation in the presence of oxygen and visible light [67] |

**Table 2.** *Cont.*

| Year | Substance [$C_0$] | Source of Light | Wavelength | Type of Reactor | Degradation/Reaction Time | Catalyst | Main Findings |
|---|---|---|---|---|---|---|---|
| 2015 | KCN [100 ppm] | 700 W Xenon lamp | n.a. | Pyrex reaction cell | 100%/5 h | CeO$_2$/KLTO | CeO$_2$/KLTO enhanced the photocatalytic activity compared to a photolytic effect at 750W/m$^2$ [68] |
| 2015 | KCN [100 ppm] | 150 W Blue fluorescent lamp | >400 nm | Horizontal cylinder annular batch reactor | 100%/30 min | S-TiO$_2$ | Photocatalytic activity resulted enhanced with the addition of S, being 0.3 wt %S-TiO$_2$ the most efficient with visible light [69] |
| 2016 | KCN [150 ppm] | 150 W Blue fluorescent lamp | >400 nm | Pyrex cell reactor | 100%/60 min | Ag-Sm$_2$O$_3$ | The Ag was beneficial for the photocatalytic activity by 90% more than bare Sm$_2$O$_3$. The catalyst is useful up to 5 times cycles [70] |
| 2016 | CN$^-$ [100 ppm] | 150 W Blue fluorescent lamp | n.a. | Pyrex glass cylindrical | 100%/70 min | Pt/Ti-Al-MCM-41 | The Pt addition to Ti-Al-MCM41 resulted in 10 times more efficient the suspended TiO$_2$ photocatalytic activity [71] |
| 2018 | KCN [30 ppm] | 25 W metal halide lamp, and UV Lamp | 365 nm and 400–700 nm | Pyrex glass cylindrical | 100%/350 min | ZnO-CuPc | 0.5wt%Zn-CuPc enhanced cyanide degradation. However, the TiO$_2$ P25 still showed faster kinetic of degradation [72] |
| 2019 | NaCN [0.18 mM CN$^-$] | 300 W Xe lamp | >400 nm | Quartz batch reactor | 90%/150 min | g-C$_3$N$_4$$^-$Nanosheets | Nanotubes exhibited good photocatalytic activity, but it was the dissolved oxygen played the most important role in the oxidation of cyanide in visible light [73] |
| 2019 | KCN [10 ppm] | Xe lamp | 400–800 nm | Pyrex glass cylindrical | 89%/120 min | B-ZnO | B-ZnO enhanced photocatalytic activity compared to bare ZnO with visible light at low cyanide concentrations [74] |

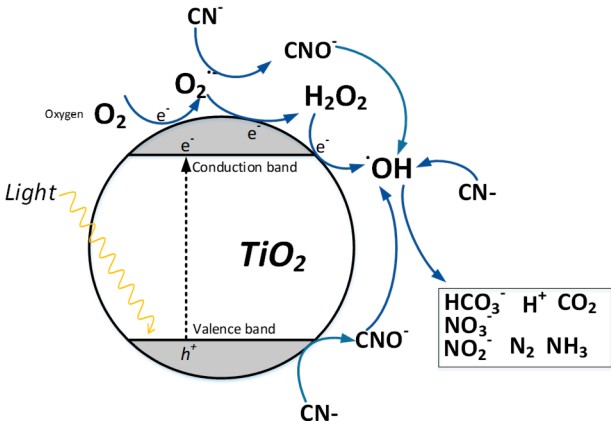

**Figure 3.** Photocatalytic scheme of free cyanide in a TiO$_2$ particle. Adapted from Reference [21].

It is well-known that the application of photocatalytic processes for chemical remediation and disinfection via hydroxyl radicals and holes oxidation has shown promising results. However, it is possible to develop other types of photodegradation without direct oxidation. Since photocatalysis is a redox process, the transformation of metallic ions and inorganic substances using the conduction band (holes) instead the valence band (electrons) can be developed. This pathway depends on the conduction and valence band energy level in the catalyst, the redox potential of the inorganic substance and the pH of the solution. In most cases, the electron transfer to a substrate is favored in the absence of oxygen in a process called reductive photocatalysis. Reductive photocatalysis has been applied to substances with oxidation states similar to CO$_2$, such as CCl$_4$, which could hardly enhance the oxidation of carbon via interaction with holes. Another example is the removal of transition metals in the solution given their multivalences when the oxidation potential is very similar to the valence band value [32].

### 4.2. Photoreduction of Metals

Photoreduction of metals was one of the first motivations for developing photocatalytic processes, and it was intended to be applied for precious metals exploitation. The main differences between photocatalysis with inorganic substances and photocatalysis with metals are (1) There are greater types of states excited by the participation of metal orbitals and ligands, (2) Conversion from one state to another is not always efficient, (3) Some excitations are achieved in the visible spectrum, (4) Heavy metals form strong spin pairing in the orbitals, generating stable and long-tripled states, and (5) The modular structure of the complexes does not allow radicals attacks, due to the organic substances. However, the excitation process by electron transfer can occur by *direct transfer* of the electron from the conduction band to substrates adsorbed on the surface of the catalyst; or *indirect reduction* by the formation of a radical product of the oxidation of organic molecules of low repulsion with the complex. Similarly, the presence of metals, such as Tl, Pb or Mn in solution has demonstrated their reductive ability of metals in solution [75].

Photoreduction processes require the absence of molecular oxygen for avoiding the formation of superoxide radical by transfer of the electron promoted to the conduction band. Furthermore, the efficiency of the reducing mechanism can be improved by the action of a sacrificing agent that is more selective for hydroxyl ($^{\bullet}$OH), hole (h$^+$), and perhydroxyl (HOO$^{\bullet}$) oxidant radicals, so that recombination is avoided and the probability of reducing other species adsorbed on the catalyst increases [76]. This reductive mechanism can decrease the oxidation state of inorganic ions and metals in solution, leading to smaller forms or their zero-valence state. This promotes the precipitation on the semiconductor surface; nevertheless, it is required that the standard potential of the level of the conduction band of the electron is sufficiently negative for generating the reduction half-reaction of the metal [77].

Figure 4 shows the different conduction and valence bands of some metal sulfides and oxides with semiconductor properties. The standard potentials (NHE) of the conduction bands (upper) and valence bands (lower) of each semiconductor are depicted. For the semiconductors with a conduction band more negative than the $H^+/H_2$ redox couple potential, the predominant mechanism is the reduction of adsorbed species; those are known as reductive semiconductors. In the opposite, for semiconductors with a balance band more positive than the $H_2O/O_2$ redox couple, are considered oxidative catalysts, generating oxidation reaction to adsorbates as its predominant mechanism.

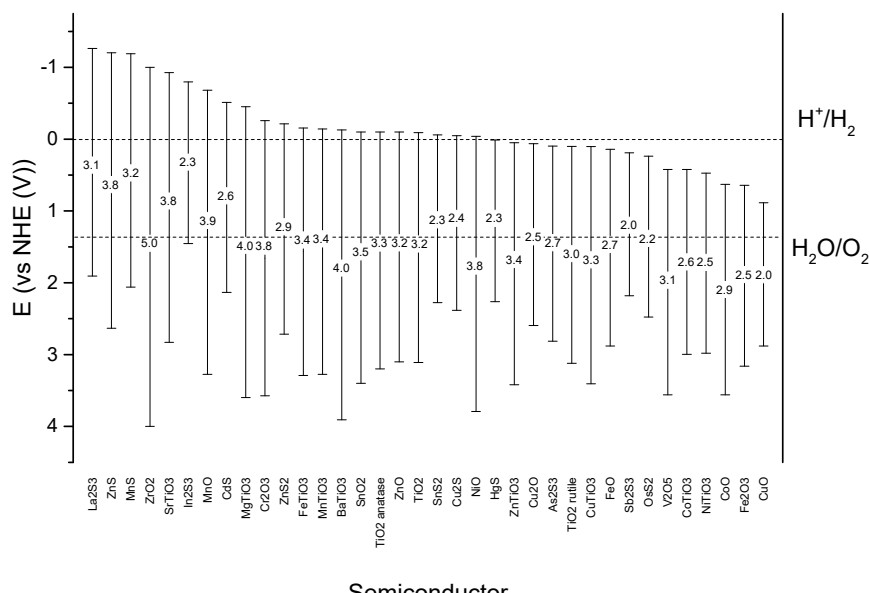

**Figure 4.** Relative position of the edges of the conduction and valence bands of some semiconductors. Adapted from References [78,79].

As it can be seen in Figure 4, the sulfides of La, Zn and Mn have conduction bands more negative than the $H^+/H_2$ redox potential, whereas metal oxides, such as Zr, Ni and $BiTiO_3$ have valence bands more positive for the $H_2O/O_2$ potential. Depending on the reduction potential of the metal or inorganic ion in solution, the most suitable semiconductor will be selected for a photocatalytic desired reaction. The most applied semiconductor catalyst in photocatalytic processes is the $TiO_2$; however, it appears in three different crystalline forms: Anatase, rutile, and brookite. From them, anatase and rutile are the most used crystalline phases in photocatalytic processes and the most commonly used in photocatalytic applications.

Few studies reported the photocatalytic reduction of metals in cyanide complexed matrices. Table 3 shows some studies related to metallic cyano-complexes using photocatalytic processes. The first evidence of solar-assisted $TiO_2$ photocatalysis studies with real mining wastewater by using As, Fe, Hg, Cu, and Zn complexed for precipitating metals, reported that three days were needed for free cyanide elimination and 17 days to achieve 99% of Hg and As removal [44]. Other studies were focused on the degradation of complex in synthetic samples and studied the degradation of Cu, Ni, Fe, Co, Pb, Cr, Au, and As cyano-complexes.

**Table 3.** Photocatalytic treatments of metallic cyano-complexes.

| Year | Matrix | Light Source | Wavelength | Type of Photoreactor | Removal/Reaction Time | Catalyst | Main Finding |
|---|---|---|---|---|---|---|---|
| 1995 | Real mining wastewater $Cu(CN)_3^{2-}$ [22 mM] $Zn(CN)_4^{2-}$ [300 mM] $Fe(CN)_6^{4-}$ [5.2 mM] Fe [29 mM] Hg [11 mM] As [16 mM] | Solar light | Solar spectrum | Dish PVC | 99% metal removals/17 days | $TiO_2$ P25 | All metal was removed with the formation of metal-hydroxides and nitrate [44] |
| 2002 | $Fe(CN)_6^{3-}$ [0.64 mM] | 150W Hg high pressure lamp | >300 nm | Pyrex batch photoreactor | 50%/350 min for SBA-15/$TiO_2$ | $TiO_2$ MCA-41, SBA-15 | The photocatalytic activity was evaluated using two different support for $TiO_2$. The porous SBA-15 resulted in better degradation of $Fe(CN)_6^{3-}$ but also for the free cyanide mineralization [48]. |
| 2003 | $Fe(CN)_6^{3-}$ [1 mM] | 4W Hg low mercury lamp and solar light | >300 nm | Cylindrical batch | 100%/1.5 h solar radiation, 77%/6 h UV Lamp | $TiO_2$ sol-gel | $TiO_2$ resulted in a better way to destroy $Fe(CN)_6^{3-}$, however resulting wastewater was rich in cyanate and incomplete oxidation was observed. Solar light exhibited better degradation rates [80]. |
| 2004 | CuCN [90 ppm $CN^-$] | 400 and 700 W halide lamp medium pressure Hg | UV | Batch cylindrical reactor | 100%/180 min | $TiO_2$ in Raschig rings support | Evaluated four different methods and the hydrothermal was the best [81]. |
| 2004 | NaCN, $Cu(CN)_3^{2-}$ [1 mM NaCN], [10 mM $Cu(CN)_3^{2-}$] | 100 W high pressure Hg lamp | 228–420 nm | Batch annular reactor bench scale. | 100%/150 min | $TiO_2$ P25 | The ratio Cu:CN influences photocatalytic degradation. A 10:1 ratio was the best for the process [82] |
| 2005 | $AuCN_2^-$ [75 mg/L $AuCN_2^-$] | 150 W medium pressure Hg lamp | 365 nm | Beaker | 86%/240 min | $TiO_2$/L | The recovery of free cyanide is made adding methanol as $^\bullet OH$ acceptor. Thus, oxidation of $CN^-$ to $CNO^-$ is avoided. The cyano-complex $AuCN_2^-$ is the electron acceptor and $Au^0$ is deposited on the $TiO_2$ particles [83] |

**Table 3.** *Cont.*

| Year | Matrix | Light Source | Wavelength | Type of Photoreactor | Removal/Reaction Time | Catalyst | Main Finding |
|---|---|---|---|---|---|---|---|
| 2005 | $[Fe(CN)_6]^{3-}$ and $[Fe(CN)_6]^{4-}$ [100 ppm $CN^-$ equivalent] | 150 W Hg medium pressure | >320 nm | Beaker | 70%/240 min | $TiO_2$ P25, $TiO_2/SiO_2$ prepared by sol-gel and hydrothermal method. | The maximum degradation was about 70% of the cyano-complex. It requires additional treatment. Iron complexes contaminated the semiconductor [84] |
| 2005 | KCN, $K_3(Fe(CN)_6)$, $KAu(CN)_2$ [3.85 mM KCN; 0.64 mM $K_3(Fe(CN)_6)$; 0.38 mM $KAu(CN)_2$] | 150 W Hg medium pressure | 365 nm | Beaker | n.d. | $TiO_2/GrSiO_2$, $TiO_2/SBA$-15. | Different methods of support were evaluated, 60% of $TiO_2/SBA$-15 performed better for iron-complex degradation [85] |
| 2008 | $CNO^-$ [0.5 mM] Fe(IV) [1 mM] | Spectro line UV-A lamp | 365 nm | Beaker | 80%/120 min | $TiO_2$ P25 Degussa | There is an enhancement in the cyanate degradation related to the presence of ferrate [65] |
| 2009 | Real Wastewater from Energy Plant | UVA UVC | 200–280; 320–400 nm | Pilot photoreactor | 100%/15 min | $FeSO_4$, $H_2O_2$ | Although the study demonstrates the ability of a pilot plant for cyanide degradation, it only is evaluated the degradation of free cyanide and not of its complexes [57] |
| 2013 | KCN, $Co(CN)_6^{3-}$, $Ni(CN)_4^{2-}$ [100 µM] | 15 W Hg low-pressure lamp. | n.d. | Cylindrical borosilicate reactor | Ni:90%/180 min, Co: 30%/180 min | $TiO_2$ P25 suspension | Nickel removal was shown to be achievable by photocatalysis; however, cobalt removal is more challenging [86] |
| 2013 | KCN, Co, Pb, Cr [100 ppm $CN^-$, Co, Pb, Cr] | Blacklight lamp and blue light | 365 nm | Annular photoreactor | 100%/180 min | $TiO_2/SiO_2$ sol-gel. | Synthesized photocatalyst could degrade free cyanide and dissolved Co, Pb, Cr. However, the evaluation of metal photo-removal was not done in the presence of cyanide [87] |
| 2018 | $Fe(CN)_6^{3-}$ [100 ppm] | 30 W UV-LED | 300–400 nm | Mini CPC UVLED photoreactor | 70%/20 min | $TiO_2$ P25 | Using UV-LED at 30W/m² in a mini CPC resulted better for recovery of cyanide instead remediation [88] |
| 2018 | $Fe(CN)_6^{3-}$ [100 ppm] | 5W UV-LED | 300–400 nm | UV baffled flat plate reactor | 60%/90 min | $TiO_2$ P25 | Configuration resulted useful for light harvesting, but it is required more UV Power since the complex was not complete degraded [89] |

Generally, the main characteristic of these photocatalytic treatments is the use of UV lamps with a high irradiation capacity (about 150, 400 or 700 W) and in some cases metals are recovered by reducing them to their zero valence state [37–41,46–48,50,54,59,62,63,66,81–86,90]. Nevertheless, photocatalytic processes for stable metallic cyano-complexes destruction are not yet fully effective for the treatment of mining wastewater, due to the presence of metal re-oxidation-redissolution and photocatalyst poisoning by deposition.

It is known the role of chemisorbed oxygen in photooxidation reactions. The $TiO_2$ chemistry depends on the $O_2$ coverage, temperature and the characteristics of the semiconductor crystalline phase. Those studies are performed using molecules, such as Ar, Kr, $N_2$, CO, $CH_4$ in order to understand their interaction with the adsorbed oxygen using a photon stimulated desorption. This technique has been used to understand and monitor photochemical processes occurring on the surface of photocatalyst [91]. Although these methods require specialized equipment, practical applications require more investment. A simpler method to understand the global mechanism is related to scavengers' addition to the bulk of the photocatalytic system.

In the case of metallic-cyanide matrices, the addition of scavengers increases the selective photoreduction of the metals (charge transfer efficiency) without oxidizing the free cyanide. Thus, several acceptors have been used as electron donors for hydroxyl, perhydroxyl, and holes. This selectivity enhancement was used to precipitate Ag from a solution of sodium cyano-argentate and sodium aurocyanide [83].

Table 4 shows examples of the main acceptors with which the mechanism studies on photocatalytic degradations in several matrices have been carried out. Compounds, such as NaF, have been used to inhibit the adsorption effect on the semiconductor particle and demonstrate the importance of the degradation reaction in bulk and not on its surface [92]. Moreover, the application of radical scavengers has been used to determine the main pathway that affects the photocatalytic degradation and to study the selectivity for certain radicals.

**Table 4.** Several radical scavenger agents used in a typical photocatalytic degradation.

| Scavengers | Compound |
|---|---|
| Holes ($h^+$) | Glucose [93]; formic acid, sulfuric acid [9,94]; sodium oxalate [95]; ammonium oxalate [96,97]; 4-methylimidozal [98]; EDTA [97,99,100]; KI [92,100,101]; $NH_4^+$ [102]; oxalic acid and methylene blue [103] |
| Hydroxyl radical ($^\bullet OH$) | t-butanol [92,96,99]; isopropyl alcohol [97]; methanol [100,104]; ethanol [101]; acetonitrile [101]; KBr [105]; terephthalic acid [106] |
| Electrons on the conduction band ($e^-$) | $Fe^{3+}$, $Cu^{2+}$, $Ag^+$ [106,107]; $AgNO_3$ [96]; $Cr^{6+}$ [95]; $KIO_3$ [102], $(S_2O_8)^{2-}$ [92] |
| Superoxide radical ($O_2^{\bullet -}$) | Benzoquinone [96,97] |

### 4.3. Application of Traditional Photoreactors and LEDs in Mining Wastewater

Sunlight and UV lamps (as natural and artificial photon source, respectively) have been used directly on photocatalytic processes with several reactor geometries, such as compound parabolic concentrators (CPC), flat plates, spinning-discs, submerged lamps and microchannel among other systems, in order to harvest light energy [108]. Nevertheless, available sunlight radiation is very variable and depends on the weather condition, geography, time of the day and year. Reversely, UV lamps as a source of UV photons for photocatalysis make this process more controllable, and it is a better alternative for fine photochemistry. The development of semiconductors for being irradiated by UV LED is a trending research topic and a promising source of photons. LED lamps have been replacing traditional incandescent halide and fluorescent mercury lamps for a wide variety of applications. The advantages of LEDs are not only the small geometry, versatility, and robustness but also a more prolonged time life, a high electrical efficiency (more than 60%) and a capability for generating radiation at a particular wavelength [109–111]. This type of artificial light has been used for water purification, sterilization, protective coatings and photo sensors from the near visible, UV to the IR spectrum range [112,113]. Table 5 shows the application of this type of diodes for photocatalytic processes as an alternative to the treatment of other organic matrices. There is only a study that reported a successful application for removal of free cyanide [21]; therefore, it can be an emerging solution for the issue of availability of UV radiation from the sun by coupling with another renewable source of energy instead using direct sunlight or traditional incandescent lamps.

**Table 5.** LED emerging photoreactors.

| Year | Description | Main Findings |
|---|---|---|
| 2013 | Phenol photodegradation using batch UV LED at 375 nm. | It is reported that UV LEDs at 800 mW are 100 times more efficient in comparison with 12 and 16 W fluorescent UV Lamps [114]. |
| 2013 | Drinking water potabilization using UV LED at 365 nm | Natural organic matter and emerging pollutants were removed from drinking water. It is concluded that the photoreactor design with this type of light is more critical than the catalyst load [111]. |
| 2014 | Used in the dyes photodegradation, organic matter of air and water. | Proved the capability of organic matter using this type of light, which is better in term of the photoreactor size, energy consumption [110]. |
| 2014 | Oxytetracycline and 17-$\alpha$-etinil estradiol as agriculture antibiotic degradation using UV LED light. | Reached a 100% degradation of total organic carbon with cumulative energy of about 12.5 kJ/L [115]. |
| 2014 | Acetonitrile degradation in Green blue and red LED photoreactor with C-N $TiO_2$. | Degradation of about 100% was achieved in 2 h using low power (3 W) LEDs [116]. |
| 2014 | Chromium photoreduction using CdS and $TiO_2$ with white LED photoreactor. | The removal of chromium was about 93% in 240 min of reaction [117]. |
| 2014 | Methyl orange degradation modeling applying Controlled Periodic Illumination in a UV LED photoreactor. | It was found that Langmuir-Hinshelwood kinetics do not describe well the photoreactor operating at Controlled Periodic Illumination. Novel mathematic modeling is required for pulsed photoreactors [118]. |
| 2014 | Selective photocatalytic reduction of nitrobenzene carried out by UV LED light. | The transformation of nitrobenzene to aniline was achieved using ethanol as the electron donor with 100% of conversion [119]. |
| 2014 | Evaluation of nitro-aromatic compounds using CdS as the catalyst. | The photoreactor uses a visible LED to enhance photoreduction of amines using methanol and isopropanol as electron donors (hole scavenger). The conversion was about 90% with a selectivity of about 71% [120]. |
| 2014 | A CFD simulation experimental and validation of a UV LED photoreactor for *Escherichia coli* disinfection. | The CFD established the best amount of irradiation, flowrate and photoreactor dimension in which best photo absorption is achieved for *E. coli* disinfection [121]. |
| 2014 | Methyl orange degradation under Controlled Periodic Illumination with a UV LED. | The Controlled Periodic Illumination demonstrated being more critical in the photonic efficiency when the ON-OFF period is closer to the characteristic time of the reaction. Also proposes photo-reductive degradation instead of a photooxidation mechanism [122]. |
| 2014 | Methyl ketone degradation using UV-vis LED with supported $TiO_2$ in alveolar foam. | The removal was 100% of methyl ketone in 600 min of reaction using 56 LEDs [123]. |

**Table 5.** *Cont.*

| Year | Description | Main Findings |
|------|-------------|---------------|
| 2014 | Photoreactor using graphene oxide ZnO for methylene blue degradation. | Degradation of 100% of Methylene blue is achieved in 150 min using UV-A LEDs. Graphene oxide resulted in photocatalytic degradation enhancement than Degussa P25 [124]. |
| 2014 | Evaluation of FeFNS-TiO$_2$ activated by LED in pesticides mineralization. | Degradation of 90% achieved in 100 min of reaction [125] |
| 2015 | *E. coli* disinfection modeling in a LED photoreactor | Bacteria deactivation achieved in 120 min using TiO$_2$ in an annular LED UV-A photoreactor [126]. |
| 2015 | Direct Red 23 degradation in a continuous UV LED photoreactor assisted with S$_2$O$_8{}^{2-}$ | Complete oxidation of Direct Red 23 is done in homogeneous photocatalysis and 72 UV-LED units [127]. |
| 2015 | Uses a photoreactor with UV-A LED for phenol and plywood mill wastewater treatment. | Demonstrated a photocatalytic degradation of phenol about >90% in 13 min and total removal of tannic acid in plywood mill wastewater in 43 min [128]. |
| 2015 | Free cyanide degradation by the oxidative pathway in synthetic wastewater. | Demonstrated the free cyanide degradation in more than 10 h, using LEDs at UVA, UVB, and UVC. The last one was the most effective in photooxidation [21]. |
| 2017 | Methylene blue degradation in a mini-CPC photoreactor. | Evaluated two systems in a coupled mini CPC and a traditional beaker with external UV-A LED illumination. Demonstrated the capability of the mini CPC in harvesting LED Light in the degradation [129]. |
| 2018 | CFD simulation to enhance LED light utilization and evaluation in iron cyano-metalic complexes. | Demonstrated the utilization of a baffled plat plate photoreactor is useful for UV-LED light harvesting [89]. |
| 2018 | Iron cyanocomplexes degraded in anoxic conditions using a mini-CPC UV LED photoreactor | Achieved the photoreduction of iron and free cyanide liberation as a strategy of recovery instead remediation for this iron cyanocomplex [88]. |

## 5. Conclusions

In this review, the state-of-the-art in the application of photocatalytic processes for the decontamination of synthetic and real cyanide wastewaters was presented. Photocatalytic processes can be effective for removing free cyanide content via oxidative pathways. Complexed cyano-metallic compounds are less studied in photoreactors, and usually, it requires the modification of selectivity by applying electron donors as scavengers of unwanted radicals in order to enhance charge transfer to the cyano-complex. The metal removal from inorganic cyanide matrices using photocatalytic processes has been explored, and the direct metal reduction on the conduction band appears to be the main mechanism as an electron acceptor at the conduction band. The use of unconventional UV LED lamps represents a growing area for development of photoreactors. Likewise, little evidence has been found of the treatment of metallic cyano-complexes from mining activities by using this type of UV source, and the existing applications are not aimed at improving the use of photons in the illuminated area. Although the evidence shows UV-vis LED application for other types of organic compounds, the knowledge about its use for the elimination or treatment of inorganic substances is still scarce. As far as it is known, none of the studies has compared the performance of the processes between different types of radiation sources at different wavelengths for cyanide wastewater treatment using UV LED.

**Author Contributions:** L.A.B.-B. compiled the literature references for photocatalytic treatments of gold mining cyanide wastewater. F.M.-M., A.H.-R., J.A.C.-M., C.F.B.-L. and L.R. restructured the information and contributed to the design, analysis and edition of the manuscript. All authors discussed the results and commented on the manuscript.

**Funding:** This research was funded by Colciencias (GRANT No. 1106-669-45250).

**Acknowledgments:** The authors are grateful to Universidad del Valle and Colciencias for the financial support to produce this work (GRANT 1106-669-45250. Recuperación de oro y tratamiento de aguas residuales cianuradas en la industria aurífera de la región pacífico Colombiana).

**Conflicts of Interest:** The authors declare no conflict of interest.

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
