# Peer review of "Recent Developments in the Photocatalytic Treatment of Cyanide Wastewater: An Approach to Remediation and Recovery of Metals"

_processes, doi:10.3390/pr7040225_

Round 1
Reviewer 1 Report
The authors in this review attempt to review the recent developments in photocatalytic treatments of cyanide wastewater. This work is a decent attempt at summarizing past relevant literature. The main critique of the manuscript is that many of the studies that the refer to only go as recent as 2015 (refer to Table 2 and 3). The literature has many more examples of the processes that the reviewers refer. Further, there is a lot left to be desired as a reader. Specifically, as a reader, I am not left with a clear picture about the new research directions, current challenges and opportunitie. Therefore, I recommend the manuscript to be revised to address the concerns mentioned above prior to publication. Some other minor comments:
1. In the abstract, the statement, “ Photocatalysis is an advanced oxidation process (AOP) able to generate a photoreaction in the solid surface of a semiconductor activated by light.” is likely incorrect. I know why the authors are writing the statement in order to link the use of photocatalysis for an advanced oxidation process. However, the statement is misleading in the correct form. Photocatalysis is a much broader term that what the authors wish to convey.
2. The statement “The mining wastewater is well known to be the predominant cause of pollution problems in surface water bodies (lakes and rivers); problems such as death due to poisoning, lead poisoning, cancer due to chromium, blindness and congenital malformations, are attributed to the contamination of surface and underground water sources.” Can be split into two sentences. In the current form, it is difficult for a reader to follow.
3. It is unclear from the authors’ description why the relation between edges of conduction band and valence band and H+/H2 potential is crucial for this topic of interest. The authors may want to try to explain this clearly.
Author Response
Please find in the attached file, the responses for the Reviewer's comments.

Reviewer 2 Report
The manuscript gives an overview regarding the developments in wastewater treatment coming from the cyanide method for gold extraction. The production of huge amount of wastewater needs the development of appropriate methods to clean the water and recover the metals. Photocatalysis seems a good candidate for wastewater treatment, both for oxidation of the organic ligands and reduction of the metals. This technique still needs to be further studied, in order to find the best operating conditions and to be applied for industrial production.
The authors well explain the problem and gives a lot of references regarding existing procedures that have been developed before, explicitating the experimental conditions. They are aware of what have been done so far and what needs to be studied.
I would propose some minor revisions to better clarify some concepts in the manuscript:
- in the introduction they don't ever talk about photocatalytic reduction of metals. They only describe the AOPs method to eliminate the toxic compounds through oxidizing species that are generated in the photocatalytic process. I would suggest to say in few words what can be done with the metals also in this paragraph, otherwise it seems that the metals are just left in the wastewater and that after evaporation and degradation, the metals remain in the soil, without any consequences.
- in the production and characterization of cyanide wastewater, raw 78 they refer to gold and silver extraction. Until that moment they only refer to gold, so I would propose to delete silver from this raw, otherwise explain what they do with silver.
- in Figure 1 it's very hard to find the words they explicit in the text: especially, I'm still looking for the word photocatalysis. Please, highlight the words that you want the reader to see.
- Figure 3 seems the same concept as shown in Figure 2. The only difference is that this is applied to the specific case of free cyanide with TiO2. I would use only one figure to describe the concept.
- All the tables that report the "main findings" should report this column larger than it is now, otherwise we have three pages of this table, that sometimes does not help to stay focus in the reading process of the paper.
- in the conclusions, why don't they refer anymore to the photocatalytic reduction of metals? Again, they only discuss the oxidative process and they don't say anything about what can be done with the photocatalysis for the removal of metals from the wastewater.
Author Response
Please find the responses in attached file.

Reviewer 3 Report
This work is a review on the current status of the applications of photo-catalytic processes to remove metals from gold extraction and its precedence in the literature. The manuscript is well-written and its structure follows a logical pattern. The report is accurate, inclusive and up-to-date and the scientific information has been conveyed with no bias and with proper and professional care. As such, I recommend the publication of the work after considering my minor comment.
When discussing the TiO2 as one of the major photo-catalytically active materials, it is not specified which polymorph of TiO2 is being discussed. TiO2 has three main crystal structures two of which are well-recognized for their catalytic functionality; namely, anatase and rutile. The band gaps of these two materials are however significantly different. As this is a review article, it is a good idea to mention such minor details in the manuscript itself rather than referring the readership to the secondary literature. As such, I request from the authors to include the band gaps and the types of the polymorphs of the TiO2 mentioned in the work if/when available in the original paper from which they are reporting.
There is a very interesting article in the literature regarding the photo-catalytic activity of TiO2 and the use of small molecules to probe its surface. I recommend for the authors to read this paper and include its information regarding the photo-catalytic activity of TiO2 in their manuscript should they deem the work necessary/relevant. The article can be found here:
Phys. Chem. Chem. Phys. 2014 Feb 14;16(6):2338-46.
Author Response
Please find the responses for the comments in the attached file.

Round 2
Reviewer 1 Report
The authors in this revision have improved the manuscript. They have still not answered my main critique of the manuscript. Many of the studies that the refer to only go as recent as 2015 (refer to Table 2 and 3). The literature has many more examples of the processes that the reviewers refer. I would recommend updating these tables (if not an explanation why) at least before acceptance.
Author Response
The answers to the reviewer's comments can be found in the attached file.
